# Intra-Cystic (In Situ) Mucoepidermoid Carcinoma: A Clinico-Pathological Study of 14 Cases

**DOI:** 10.3390/jcm9041157

**Published:** 2020-04-18

**Authors:** Saverio Capodiferro, Giuseppe Ingravallo, Luisa Limongelli, Mauro Giuseppe Mastropasqua, Angela Tempesta, Gianfranco Favia, Eugenio Maiorano

**Affiliations:** 1Department of Interdisciplinary Medicine – Section of Odontostomatology, University of Bari Aldo Moro, Italy, Piazza G. Cesare, 11, 70124 Bari, Italy; capodiferro.saverio@gmail.com (S.C.); lululimongelli@gmail.com (L.L.); angelatempesta1989@gmail.com (A.T.); gianfranco.favia@uniba.it (G.F.); 2Department of Emergency and Organ Transplantation – Section of Pathological Anatomy, University of Bari Aldo Moro, Italy, Piazza G. Cesare, 11, 70124 Bari, Italy; mauro.mastropasqua@uniba.it (M.G.M.); eugenio.maiorano@uniba.it (E.M.)

**Keywords:** salivary glands, minor salivary glands, salivary gland carcinoma, mucoepidermoid carcinoma, in situ carcinoma, intra-cystic carcinoma

## Abstract

Aims: To report on the clinico-pathological features of a series of 14 intra-oral mucoepidermoid carcinomas showing exclusive intra-cystic growth.

Materials and methods: All mucoepidermoid carcinomas diagnosed in the period 1990–2012 were retrieved; the original histological preparations were reviewed to confirm the diagnosis and from selected cases, showing exclusive intra-cystic neoplastic components, additional sections were cut at three subsequent 200 m intervals and stained with Hematoxylin–Eosin, PAS, Mucicarmine and Alcian Blue, to possibly identify tumor invasion of the adjacent tissues, which could have been overlooked in the original histological preparations. Additionally, pertinent findings collected from the clinical charts and follow-up data were analyzed.

Results: We identified 14 intraoral mucoepidermoid carcinomas treated by conservative surgery and with a minimum follow up of five years. The neoplasms were located in the hard palate (nine cases), the soft palate (two), the cheek (two) and the retromolar trigone (one). In all instances, histological examination revealed the presence of a single cystic space, containing clusters of columnar, intermediate, epidermoid, clear and mucous-producing cells, the latter exhibiting distinct intra-cytoplasmic mucin production, as confirmed by PAS, Mucicarmine and Alcian Blue stains. The cysts were entirely circumscribed by fibrous connective tissue, and no solid areas or infiltrating tumor cell clusters were detected. Conservative surgical resection was performed in all cases, and no recurrences or nodal metastases were observed during follow up.

Conclusions: Mucoepidermoid carcinomas showing prominent (>20%) intra-cystic proliferation currently are considered low-grade tumors. In addition, we also unveil the possibility that mucoepidermoid carcinomas, at least in their early growth phase, may display an exclusive intra-cystic component and might be considered as in situ carcinomas, unable to infiltrate adjacent tissues and metastasize.

## 1. Introduction

Mucoepidermoid carcinoma (MEC) was firstly described by Volkmann in 1895; subsequently, Stewart et al. (1945) defined such lesion as a “mucoepidermoid tumor”, and identified tumors with “relatively favorable” and “highly unfavorable” clinical outcomes. Later on, Jakobsson et al. and many other authors [1,2,3,4] proposed to separate MECs into low, intermediate and high grades, based on the relative proportion of cell types, a distinction that still persisted in the WHO classification of tumors of 2017 [5]. MEC is one of the most common salivary gland malignancies, showing distinctive morphological features, such as mucous, intermediate and epidermoid cells in variable proportions [5,6,7]. Less than half of the cases arise in minor salivary glands, the palate being the most common intra-oral localization of MEC [8,9,10,11,12]. The architectural configuration of MEC may vary, but a cystic component is commonly present and may sometimes predominate [5,6,13,14,15]. Nevertheless, most MECs also show a solid growth pattern and infiltration of adjacent structures [16,17].

Though considered a tumor with low malignant potential in most instances, about 10% of the patients affected by MEC experience tumor-related death [10,11,18,19]. In this regard, MECs located in the submandibular gland and those showing a high histopathologic grade are considered more aggressive [8,9,10,20,21]. It should be noticed that the greater extension of the intra-cystic component correlates with lower grade of MEC, and therefore, this tumor characteristic per se may influence the clinical outcome [5,6,7,8,18,19,22].

Based on these premises, while retrospectively re-evaluating all MECs examined in the period 1990–2012, we focused our attention on those cases showing prevalent/exclusive intra-cystic components to further characterize their relevance in the clinic-pathological presentations and clinical outcomes of the affected patients.

## 2. Materials and Methods

All cases diagnosed as MEC at our institution during the years 1990–2012 were retrieved from the files of the Section of Pathological Anatomy of the University of Bari Aldo Moro, along with the pertinent clinical charts and follow-up data updated as of January 2019. All cases were fixed in 10% neutral buffered formalin, embedded in paraffin and routinely stained with Hematoxylin–Eosin; Periodic Acid–Shiff (PAS), with and without diastase pre-treatment; Mucicarmine; and Alcian Blue. The original histological preparations were reviewed to confirm the diagnoses, based on the occurrence of the distinct cell types (squamoid, mucous-producing and intermediate cells) that characterize MEC [5]. Additional sections at 3 subsequent 200 µm intervals were cut of selected cases showing exclusive tumoral intra-cystic components and stained with the above procedures, to possibly identify tumor invasion of the adjacent tissues that could have been overlooked in the original histological preparations. 

## 3. Results

During the observational period, 128 MECs were identified, 82 involving the major and 46 the minor salivary glands; among these, 14 showed an exclusive intra-cystic tumoral component in the absence of infiltration of the adjacent tissues, as confirmed by the evaluation of additional cutting levels, the salient clinico-pathological features of which are reported in Table 1. Among such patients there were three males and 11 females, with a median age of 36.8 years; nine MECs involved the hard palate, two cases the soft palate, two the cheek mucosa and one case the retromolar trigone. In all instances, the neoplasms appeared as intra-oral nodules (Figure 1), sometimes with slight erosion/ulceration of the surface epithelium; and showed painless, slow growth and hard consistency, without evident infiltration of the adjacent soft and hard tissues, as confirmed by MR (conventional acronym for Magnetic Resonance) and CT scans. The tumor dimensions were relatively small, with a minimum clinical diameter of 0.5 cm up to a maximum of 1.8 cm. No loco-regional node involvement was detectable by clinical inspection or imaging techniques in any instance. All patients underwent conservative surgical excision with a rim of normal tissue. It should be emphasized that all the tumors of this cohort were localized in minor salivary glands and we were unable to identify “pure” intra-cystic (in situ) MEC in major glands.

Gross examination disclosed well defined cystic lesions, and microscopically, at scanning magnification, a single cystic space was detectable in all samples, showing parietal proliferation of clusters of epithelial cells with a focal cribriform growth pattern (Figure 2). The central part of the cyst was filled with proteinaceous material and cholesterol crystals, while a distinct and complete rim of collagenous stroma separated the cyst from the surface epithelium and from adjacent lobules of mucous salivary glands. The clusters of epithelial proliferation (Figure 3) were composed by small columnar and intermediate cells, cells with prominent cytoplasmic clearing and marginated nuclei, scattered flat to polygonal cells showing epidermoid differentiation and a reduced number of large mucous-producing cells with multivacuolated cytoplasm. The latter were better highlighted with Alcian Blue (Figure 4) and Mucicarmine stains and also showed PAS-positivity, which was partly abolished after diastase treatment. Occasionally, smaller cystic spaces with cribriform appearance were evident within the neoplastic epithelial clusters, which were lined by cuboidal to columnar cells. Nuclear pleomorphism was minimal, as was mitotic activity (<1/10 high power fields), while inflammatory infiltration, necrosis and perineural invasion were undetectable; additionally, tumor-free margins (> 1 mm) were assessed in all cases. Patients had been followed-up for a minimum of five years (range: 62–120 mo.; median: 68 mo.) and had remained without evidence of disease up to January 2019.

## 4. Discussion

Salivary gland carcinomas represent about 5% of all head and neck carcinomas and 0.5% of all malignancies [5,8,9,10,23,24,25], with an incidence of 1.1/100000 per year in the Caucasian population [9,25,26,27], and have been classified into 20 different types by the World Health Organization in 2017 [5].

MEC is the most common malignant tumor of the salivary glands (12%–29%) in children and young adults [9,25,28,29], and according to some authors, the most common malignancy in minor salivary glands [8,9,10,11,12,13,20]; its peak incidence is between the third and sixth decade, with predilection for females [25,28,29]. As confirmed by the results of the present study, the palate remains the most common site for MECs occurring in minor salivary glands, while they less frequently occur in the retromolar area, the floor of the mouth, the buccal mucosa, the lips and the tongue [3,5,18,23].

The cases reported herein showed the distinctive morphological features of “classical” MEC [5,30,31]; i.e., an epithelial tumor composed of intermediate, epidermoid, mucous-producing and clear cells, arranged in irregular clusters of variable size, but at variance with conventional MEC, no foci of stromal invasion were detected and the neoplastic proliferation manifested an exclusive intra-cystic growth. It is our opinion that the presence of a continuous rim of collagen around the neoplastic proliferation better testifies its in situ nature, and the presence of minimal stromal invasion or isolated tumor cell clusters should be accurately excluded, by examining the sample at multiple cutting levels. Immunohistochemical stains for myoepithelial (e.g., smooth muscle actin, calponin, smooth muscle myosin heavy chain) or basal cell markers (e.g., cytokeratin 14, p63) could possibly help to clarify this issue. Nevertheless, as for breast in situ carcinomas, residual ductal myoepithelial cells are not usually present as a continuous layer; therefore, it is somewhat misleading to assess the true in situ nature of the tumor. Additionally, anti-cytokeratin 14 and p63 antibodies do not stain residual ductal cells only; variable proportions of neoplastic MEC cells are positive for these markers as well.

Traditionally, MEC is considered a tumor with low malignant potential, though cases showing local recurrence, nodal and distant metastases and tumor-related death have been repeatedly reported [8,9,14,17,19,27]. Tumor aggressiveness is strictly related to histological grade, and although there is not complete agreement on the grading systems proposed so far, a three-tiered scale considering low, intermediate and high grade MECs has been more commonly adopted and proven useful for prognostic purposes [5,6,13,25,31]. Such systems take into account the extension of the intra-cystic component, the presence of neural invasion and necrosis, the mitotic index and cellular anaplasia [12,14,15,24,31]. At this regard, the cases of the present series, while fitting into the morphological diagnosis of “conventional” MEC as to their cell components, did not show but minimal nuclear pleomorphism, and occasional, if any, mitotic figures, in the absence of perineural/bone invasion and necrosis, thereby qualifying as low grade tumors, with indolent clinical behavior. Herewith, we provide morphological evidence to postulate that a less aggressive form of MEC may be identified, as for epithelial tumors occurring in other organs (e.g., breast and prostate), which could be considered an in situ carcinoma. This novel tumor subtype, characterized by exclusive intra-cystic growth, should be, by definition, incapable of infiltrating adjacent tissues and giving raise to nodal or distant metastases, thereby being easily curable with conservative surgery. Such clinico-pathological features parallel those of in situ (lobular/ductal/papillary) carcinomas of the breast and support the concept that early identification of such neoplasms may allow less aggressive treatments.

Furthermore, based on the classical morphological features of MEC, intraductal papilloma, cystadenoma, adenosquamous carcinoma, salivary duct carcinoma and salivary gland clear cell carcinomas could be considered in the differential diagnosis [21,22,30,31,32]. The lack of any papillary growth pattern and the presence of distinct and frequently prevalent clusters of intermediate cells help to rule out intraductal papilloma and cystadenoma, respectively. Adenosquamous carcinoma and salivary duct carcinoma may closely mimic MECs, but such tumor types are devoid of intermediate cells and usually show higher degrees of cellular pleomorphism and mitotic activity. In addition, evident mucin production within the neoplastic cells contributes to excluding other types of salivary gland carcinomas with clear cells (e.g., acinic cell carcinoma, hyalinizing carcinoma), which also lack an intermediate cell population [14,22,32].

It is well known that MECs may harbor MAML2 gene fusion, but in view of the typical morphologic features of all case of the present series, we considered the assessment of the status of MAML2 would have not added much to this study. In fact, it is generally accepted that up to 75%–80% of MECs, especially low and intermediate grades, harbor gene fusions involving MAML2 [33,34,35,36]. Despite its high specificity, MAML2 testing is no longer considered a useful prognostic indicator for already diagnosed MECs, and may be avoided when the diagnosis of MEC is reached straightforwardly, based on typical morphologic features [34,35,37]. Consequently, MAML2 testing as an ancillary diagnostic tool should be reserved for MECs showing unusual histological appearances [38,39], such as the oncocytic variant of MEC, to rule out oncocytoma and oncocytic carcinoma; and the Whartin-like variant and the recently described ciliated MEC variant, to rule out benign developmental cysts and ciliated, HPV-related squamous cell carcinomas [40,41].

In addition, we would also like to emphasize that we were unable to identify MECs with exclusive intra-cystic (in situ) growth in major salivary glands, but this may be related to the higher chances of detecting such tumors at earlier growth phases when located in intra-oral sites, in view of easier accessibility to inspection and palpation. In other words, we cannot exclude that intra-cystic (in situ) MECs may be present in major salivary glands, but they possibly remain undetected for longer times and are disclosed when infiltration of adjacent tissues has already occurred.

The pathogenesis of malignant salivary gland neoplasms, as well as the occurrence of genetic and epigenetic alterations still remain unclear. However, it would be interesting to explore whether the typical chromosomal translocation t (11;19) (MECT1-MAML2), which is detected in >50% of “conventional” MEC, is already present in tumors at early stages of tumorigenesis, such as those of the present series, and whether additional genetic alterations might be responsible for further progression to frankly invasive MEC. 

In conclusion, we provide evidence that MECs with exclusive intra-cystic (in situ) components showed indolent clinical behavior, with no evidence of recurrence or metastases even after prolonged follow up, and may be more conservatively treated. Therefore, we strongly suggest adopting the designation intra-cystic (in situ) MEC in diagnostic reports to properly manage patients and avoid unnecessary over-treatments.

## Figures and Tables

**Figure 1 jcm-09-01157-f001:**
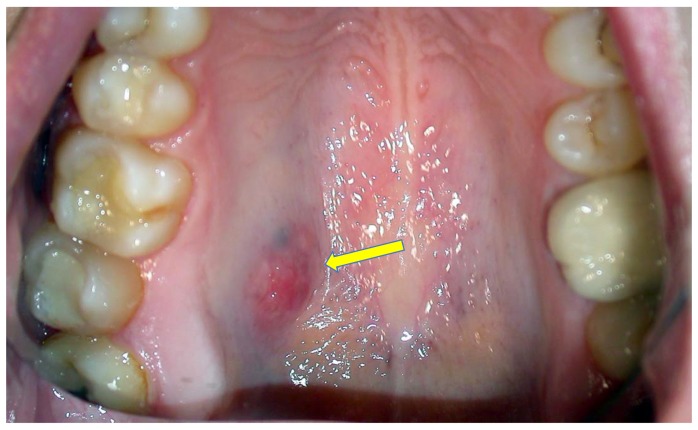
Clinical presentation MEC of the hard palate as a rather well-demarcated nodule with slight erosion of the covering mucosa.

**Figure 2 jcm-09-01157-f002:**
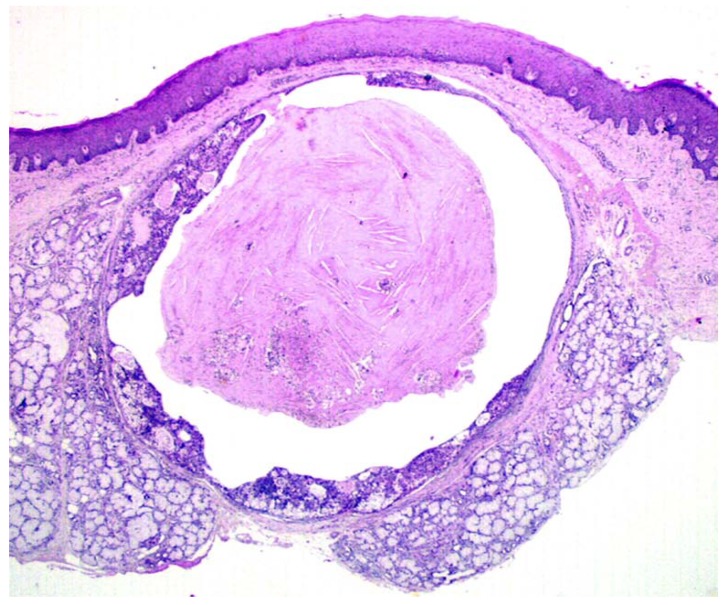
Under scanning magnification, the tumor was composed of a single cystic space, partly filled with proteinaceous material and cholesterol crystals, and showed parietal growth of epithelial cells and complete peripheral demarcation by fibrous connective tissue. (H&E, x1).

**Figure 3 jcm-09-01157-f003:**
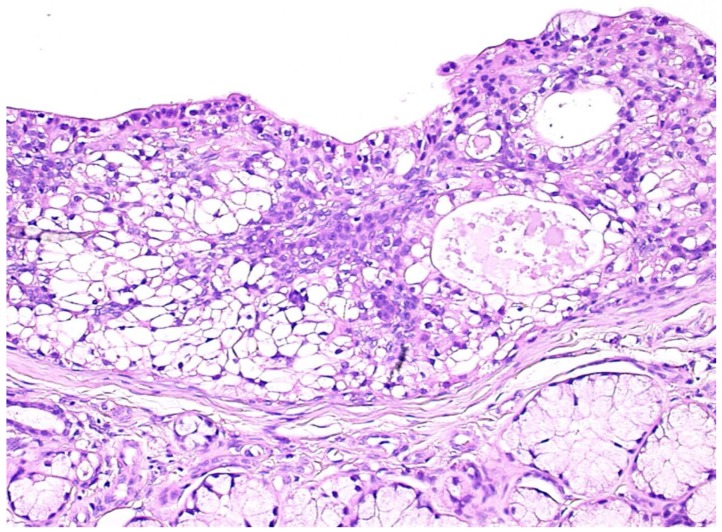
The epithelial component consisted of small columnar and intermediate cells, cells with prominent cytoplasmic clearing, rare flat to polygonal cells showing epidermoid differentiation and a reduced number of large mucous-producing cells with multivacuolated cytoplasm. (H&E, x10).

**Figure 4 jcm-09-01157-f004:**
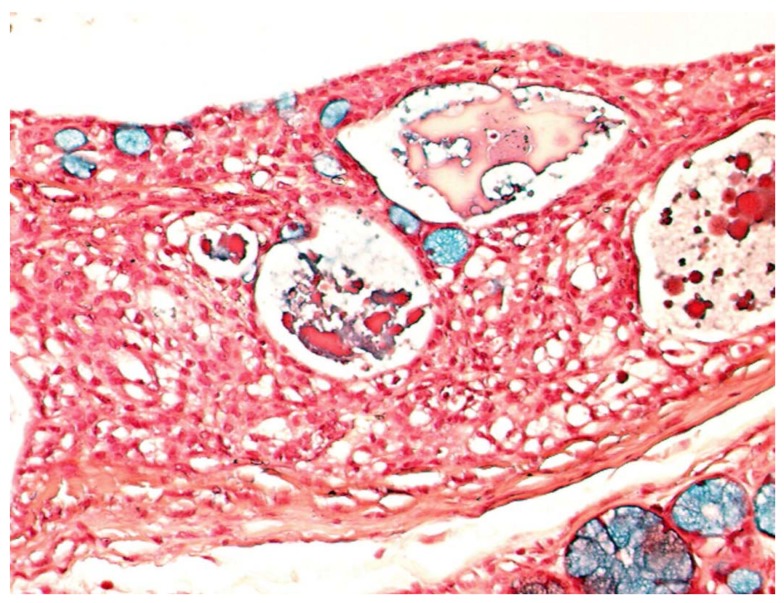
Epithelial cells with multi-vacuolated cytoplasms and marginated nuclei demonstrate consistent Alcian Blue positivity, indicating mucous production. (Alcian Blue, x20).

**Table 1 jcm-09-01157-t001:** Clinico-pathological features of the patients with intra-cystic mucoepidemoid carcinoma (all alive without evidence of disease after the specified follow-up interval).

Case #	Age	Sex	Site	Size (cm)	Follow-up (months)
1	51	F	Hard palate	0.6	66
2	26	F	Hard palate	1.8	62
3	20	M	Soft palate	1.3	88
4	25	F	Hard palate	0.7	74
5	36	F	Hard palate	0.5	84
6	35	F	Cheek	0.9	68
7	34	F	Hard Palate	1.0	74
8	41	F	Cheek	1.2	62
9	28	M	Soft palate	0.8	68
10	46	F	Hard palate	1.1	62
11	50	M	Hard palate	1.8	120
12	39	F	Hard palate	1.2	95
13	45	F	Retromolar trigone	1.6	66
14	40	F	Hard palate	1.2	68

# = conventional symbol for “number”.

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
