# Peer review of "Intra-Cystic (In Situ) Mucoepidermoid Carcinoma: A Clinico-Pathological Study of 14 Cases"

_jcm, 2020, doi:10.3390/jcm9041157_

Round 1

Reviewer 1 Report

The present paper describes a novel variant of mucoepidermoid carcinoma (MEC) of the minor salivary glands, the intracystic variant.

The paper is very interesting. Intracystic low-grade carcinomas are well known in the breast a distinctive entity. In this paper the concept has been applied to intra-oral MEC.

The study is based on 14 cases, with good clinic-pathological details and long follow-up.

The results are very interesting. It is suggested to better specified the diagnostic criteria used to diagnose the pure  intracystic carcinoma. Specifically: was a myoepithelial layer present? If present how it was evaluated? CK14 and p63 stain the basaloid cells usually present in MEC. Were calponin or smooth muscle actin applied? Was a fibrous wall present? It is suggested to propose a clear definition of “pure intracystic MEC”.

Results:”…. also, tumor-free margins were assessed in all cases.”: distance in mm from tumour to the closest margin should be reported.

Results: “Patients had been followed-up for a minimum of 5 years and had remained without evidence of disease up to January 2019”: follow-up length should be reported (range and mean).

Discussion: “In addition, we would also like to emphasize that we were unable to identify MEC with exclusive intra-cystic (in situ) growth in major salivary glands but this may be related to higher chances to detect such tumors at an earlier growth phase when located in intra-oral sites, in view of easier accessibility to inspection and palpation. In other words, we cannot exclude that intra-cystic (in situ) MEC may be present in major salivary glands but, possibly, they remain undetected for longer times and are disclosed when infiltration of adjacent tissues has already occurred.” This is a very important point, but it is not well evident in the Results section.

Discussion: a parallel witu intracystica breast carcinoma should be considered.

Discussion: a fee sentences of conclusions, focusing on the importance to diagnose this new MEC variant should be added.

Reviewer 2 Report

This retrospective study about intra-cystic MEC is very interesting and I think it is scientific significant. I have  some proposals:

  1. It will be helpful if authors show what is the frequency of this tumor. They found 14 but from how many MEC.
  2. It will be more clear if they marked somehow the nodule in Figure 1
